# Comment on Vicente-Rabaneda et al. A Proposal for a New Lung Ultrasound Score in Rheumatoid Arthritis: The Reliability of Lung Ultrasound for Rheumatoid Arthritis-Associated Interstitial Lung Disease Diagnosis. *J. Clin. Med.* 2025, *14*, 3701

**DOI:** 10.3390/jcm14228118

**Published:** 2025-11-17

**Authors:** María Agustina Gomez-Rojas, Yale Tung-Chen

**Affiliations:** 1Department of Rheumatology, Hospital Universitario La Paz, Paseo de la Castellana, 261, 28046 Madrid, Spain; mariaagustina.gomez@salud.madrid.org; 2Derpartment of Internal Medicine, Hospital Universitario La Paz, Paseo de la Castellana, 261, 28046 Madrid, Spain; 3Departament of Medicine, Facultad de Medicina, Universidad Autónoma de Madrid, 28046 Madrid, Spain

We read with great interest the article titled “A Proposal for a New Lung Ultrasound Score in Rheumatoid Arthritis: The Reliability of Lung Ultrasound for Rheumatoid Arthritis-Associated Interstitial Lung Disease Diagnosis” [1], and we thank the authors for highlighting the value of bedside lung ultrasound (LUS) in patients with rheumatoid arthritis (RA) and other rheumatologic conditions. However, we would like to offer some methodological and conceptual considerations that we believe are essential to accurately interpret the study’s findings and their applicability to clinical practice.

First, we believe that combining a monocentric case series with a systematic review in the same manuscript may lead to confusion, as both types of evidence have distinct levels of internal and external validity. While a small case series is inherently exploratory, with high risk of selection bias and limited generalizability, a systematic review—when rigorously conducted—offers stronger methodological robustness and broader external validity. The integration of both does not strengthen the original study’s conclusions; on the contrary, it may inadvertently suggest a level of evidence that is not supported, especially considering that the studies included in the review differ markedly in terms of patient selection and LUS methodology.

We also agree with the authors’ own recognition that the small sample size reduces the precision of reliability estimates and may not capture the clinical and ultrasonographic variability present in the general RA population with lung involvement. The monocentric design further limits the generalizability of the findings, as differences in patient characteristics, equipment, and operator expertise across centers may influence results. Moreover, the consecutive inclusion of patients without stratifying for the severity of lung involvement could skew the outcomes toward the most prevalent patterns within that specific cohort. This potentially under- or overestimates the technique’s reliability in milder or more advanced cases. Importantly, the exclusion of asymptomatic patients may hinder early detection of subclinical lung disease. Previous studies have shown that over 60% of newly diagnosed RA patients may exhibit pulmonary abnormalities on imaging or functional tests even in the absence of respiratory symptoms. Therefore, limiting inclusion to symptomatic individuals introduces a selection bias and affects the study’s external validity [2].

In addition, it would be valuable to report whether patients were smokers [3] and which treatments they were receiving, as both smoking and medications such as methotrexate, leflunomide, or biologics may influence lung findings. While recent studies suggest that methotrexate does not significantly increase the risk of interstitial lung disease (ILD), its historical association with pulmonary toxicity warrants documentation. Leflunomide may worsen ILD in patients with severely impaired lung function, and the pulmonary impact of biologics remains controversial. Documenting and analyzing these exposures are essential for accurate interpretation [4].

Regarding LUS findings, it is important to note that B-lines and pleural irregularities are not specific to ILD. They may also be seen in conditions such as heart failure, pneumonia, acute respiratory distress syndrome, and post-inflammatory pleural changes. Thus, their presence must be clinically contextualized and, when possible, corroborated using complementary diagnostic methods [5].

We also acknowledge the innovation that LUS represents in rheumatologic practice—its low cost and radiation-free nature make it an attractive diagnostic adjunct. Nonetheless, the fact that all scans were performed by highly experienced sonographers may limit the external validity of the results. Studies involving operators with varying levels of expertise, including rheumatology residents or general practitioners, have reported lower interobserver reliability, underscoring the critical role of operator training in reproducibility [6].

Finally, the recent International Guidelines and Consensus on Lung Ultrasound emphasize the urgent need for objective criteria to assess pleural line irregularity/thickening and to differentiate micro- from macro-subpleural consolidations in terms of both size and ultrasonographic appearance [7]. Therefore, while LUS is a promising technique, it may become an important clinical tool to be integrated with HRCT in the screening and follow-up of ILD.

In conclusion, while we share the authors’ enthusiasm for the potential implementation of LUS as a diagnostic tool for RA-ILD, it is essential to recognize not only the limitations of the study but also those inherent to LUS as a standalone diagnostic modality in this complex clinical context.

## Data Availability

The authors confirm that the data supporting the findings of this study are available from the corresponding author, upon reasonable request.

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
