# Peer review of "Comment on Vicente-Rabaneda et al. A Proposal for a New Lung Ultrasound Score in Rheumatoid Arthritis: The Reliability of Lung Ultrasound for Rheumatoid Arthritis-Associated Interstitial Lung Disease Diagnosis. J. Clin. Med. 2025, 14, 3701"

_jcm, 2025, doi:10.3390/jcm14228118_

Round 1

Reviewer 1 Report

Comments and Suggestions for Authors

In their commentary, the authors address several aspects of the original article by Di Matteo et al, which are important to consider regarding the value of lung ultrasound (LUS) in the diagnosis of rheumatoid arthritis (RA)-associated interstitial lung disease (ILD). Their arguments and considerations are clearly reported and well justified. The authors are advised to consider the further inclusion and/or elaboration of the following two points, which could provide additional value in this commentary:
1) The differential role of LUS in the screening for RA-ILD in RA patients without respiratory manifestations versus in the diagnosis of RA-ILD in RA patients with respiratory manifestations. While chest high resolution computed tomography (HRCT) is an irreplaceable imaging modality for the diagnosis of RA-ILD in suspect RA cases, LUS could have a more important role in the screening of asymptomatic RA patients for RA-ILD and their further referral for chest HRCT if relevant abnormal findings are depicted by LUS.
2) While the presence of B-lines per se is indeed a highly non-specific LUS finding for ILD, as correctly commented in lines 76-80, the particular characteristics of the ultrasonographic interstitial syndrome could improve the specificity of B lines for ILD, especially in the right clinical context, e.g., their non-predominance in the gravity-dependent regions of the lungs, their persistence despite diuretic therapy, the coexistence of pleural line fragmentation, the absence of pleural effusions.

Reviewer 2 Report

Comments and Suggestions for Authors

Dear authors,

The manuscript is written well with clear and balanced comments on the article "A proposal for new Lung....". You have identified relevant methodological limitations of the study (small sample size, combination of monocentric case series with systematic review, operator dependency of LUS...). Especially the exclusion of asymptomatic patients which would be the most important group to study considering the nature of RA-ILD.